# Physiological TLR4 regulation in human fetal membranes as an explicative mechanism of a pathological preterm case

Corinne Belville[1], Flora Ponelle-Chachuat[1], Marion Rouzaire[1], Christelle Gross[1], Bruno Pereira[2], Denis Gallot[1,3], Vincent Sapin[1,4], Loïc Blanchon[1]*

[1]Team 'Translational approach to epithelial injury and repair', iGReD, Université Clermont Auvergne, Clermont-ferrand, France; [2]CHU Clermont-Ferrand, Biostatistics unit (DRCI) Department, clermont-ferrand, France; [3]CHU Clermont-Ferrand, Obstetrics and Gynaecology Department, Clermont-ferrand, France; [4]CHU Clermont-Ferrand, Biochemistry and Molecular Genetic Department, Clermont-Ferrand, France

**ABSTRACT** The integrity of human fetal membranes is crucial for harmonious fetal development throughout pregnancy. Their premature rupture is often the consequence of a physiological phenomenon that has been exacerbated. Beyond all the implied biological processes, inflammation is of primary importance and is qualified as 'sterile' at the end of pregnancy. In this study, complementary methylomic and transcriptomic strategies on amnion and choriodecidua explants obtained from the altered (cervix zone) and intact fetal membranes at term and before labour were used. By cross-analysing genome-wide studies strengthened by in vitro experiments, we deciphered how the expression of toll-like receptor 4 (TLR4), an actor in pathological fetal membrane rupture, is controlled. Indeed, it is differentially regulated in the altered zone and between both layers by a dual mechanism: (1) the methylation of TLR4 and miRNA promoters and (2) targeting by miRNA (let-7a-2 and miR-125b-1) acting on the 3'-UTR of TLR4. Consequently, this study demonstrates that fine regulation of TLR4 is required for sterile inflammation establishment at the end of pregnancy and that it may be dysregulated in the pathological premature rupture of membranes.

*For correspondence:
loic.blanchon@uca.fr

Competing interest: The authors declare that no competing interests exist.

## Editor's evaluation

This paper sheds light on mechanisms regulating TLR4 expression in fetal membranes.

## Introduction

Placenta and fetal membranes are extra embryonic tissues that are originally formed by trophoblastic cell differentiation. Although these organs are transitory, their integrities throughout pregnancy are essential for harmonious in utero fetal development. Protecting the fetus (which is considered a semi-allogeneic graft), fetal membranes are composed of amnion and chorion, creating a 0.5 mm thick layer surrounding the amniotic cavity. They are important for the protection of the fetus against infections ascending the genital tract and for the initiation of programmed term rupture (*Joyce et al., 2009*; *Kendal-Wright, 2007*). The site of rupture is a particular and unique zone of altered morphology (ZAM) situated around the cervix (*McLaren et al., 1999*; *McParland et al., 2003*), which, along with a zone of intact morphology (ZIM), presents specific histological and cellular characteristics (*El Khwad et al., 2006*; *Mauri et al., 2013*; *de Castro Silva et al., 2020*). Furthermore, these processes lead to

the emergence of a phenomenon called 'sterile inflammation', which occurs because of the release of numerous DAMP (damage-associated molecular pattern) molecules, such as HMGB1, the S100 protein family and HSPs (*Zindel and Kubes, 2020*), all of which provoke the release of proinflammatory cytokines (*Gomez-Lopez et al., 2014*).

Dysregulation of the fetal membrane's integrity or premature activation of inflammatory pathways could lead to preterm premature rupture of membranes (PPROM), which is usually defined when occurring before 37 weeks of gestation and encountered in 1–3% of all births (*Waters and Mercer, 2011*). This is associated with a high mortality rate and significant morbidity in newborn survivors because of fetal prematurity and maternal complications (*Goldenberg et al., 2008*). Both sterile intra-amniotic inflammation and intra-amniotic infections (Gram-negative bacteria) are the causes for PPROM pathology, whereas transcriptomic profiles could be different (*Konwar et al., 2018*; *Motomura et al., 2021*; *Musilova et al., 2015*; *Romero et al., 2006*). Nevertheless, they share common cellular actors, such as surface-expressed pattern recognition receptors, as the innate components of the immune system, including the most frequently described toll-like receptor (TLR) family (*Kawai and Akira, 2010*; *Newton and Dixit, 2012*).

To better understand the physiological and pathological rupture of membranes, the molecular study of global changes in gene expression can be accomplished using high-scale technical analyses (*Eidem et al., 2015*; *Haddad et al., 2006*; *Han et al., 2008*; *Kawai and Akira, 2010*; *Li et al., 2011*; *Lim et al., 2012*). Surprisingly, only one unique transcriptome study focused on the ZIM or ZAM regions and amnion and chorion tissues, characterising a specific differential expression in a spontaneous rupture at term with labour; in the study, differences were observed in the chorion, though not in the amnion, specifically involving biological processes, such as extracellular matrix–receptor interaction and inflammation (*Nhan-Chang et al., 2010*). In addition to classical direct gene regulation by transcription factors, DNA methylation is another well-known epigenetic mechanism that can interfere with the transcriptional regulation of all RNA types. This has only been investigated in one study on the amnion methylation status between labour and nonlabour explants (*Kim et al., 2009*; *Kim et al., 2013*).

The purpose of our study is to exhaustively combine and correlate methylomic and transcriptomic analyses between ZIM and ZAM in the case of 'sterile inflammation'. By cross-analysing our genomewide studies, our objectives are to explain the different levels of gene expression by comparing the amnion/choriodecidua and the ZIM/ZAM. At term without labour (to avoid contraction influence and bacteria contamination at parturition), here by classifying genes into specific biological processes, we demonstrated that toll-like receptor 4 (TLR4), which is classically involved in the recognition of Gram-negative bacteria and that triggers an inflammatory response in chorioamnionitis leading to PPROM (*Medzhitov et al., 1997*; *Poltorak et al., 1998*), was overexpressed in the ZAM choriodecidua compared with the ZAM amnion. The mechanisms implying TLR4 in the physiological or pathological rupture of membrane in case of PPROM are well known. Triggering TLR4 will lead to NF-κB activation, leading to an increase of the release of proinflammatory cytokine, concentration of matrix metalloprotease and prostaglandin, which are well established actors of fetal membrane rupture (*Robertson et al., 2020*). Furthermore, we discovered that TLR4 regulation leads to layer and zone specificity. The latter occurred because of the hypomethylation of the TLR4 gene body in the ZAM choriodecidua, whereas its weak expression in the ZAM amnion layer was a direct consequence of the action of two hypomethylated miRNAs targeting the 3-UTR of TLR4: let-7a-2 and miR-125b-1. Therefore, the physiological choriodecidual overexpression of TLR4 could be exacerbated in PPROM, leading to the enhancement of the first step of early fetal membrane rupture.

## Results
### Methylomic analysis of fetal membranes allows for defining the Gene Ontology classification for the ZAM

After identifying the differentially hypermethylated genes between the amnion and choriodecidua on the whole genome between the ZIM and ZAM (*Figure 1A*), biological process analysis was performed; this has been represented by a four-way Venn diagram (*Figure 1B*). If the hypermethylated genes were clearly lower in the ZIM (1746 genes) than the ZAM (9830 genes), the specific genes only found in the ZIM (hyper- or hypomethylated; total number 98 [i.e., 10 + 88, as illustrated by black circles]) did

**A**

| | ZIM | | ZAM | |
|---|---|---|---|---|
| | Hypermethylated Cytosines in Amnion (mA) | Hypermethylated Cytosines in Choriodecidua (mC) | Hypermethylated Cytosines in Amnion (mA) | Hypermethylated Cytosines in Choriodecidua (mC) |
| Genes Number | 322 | 1424 | 5086 | 4744 |

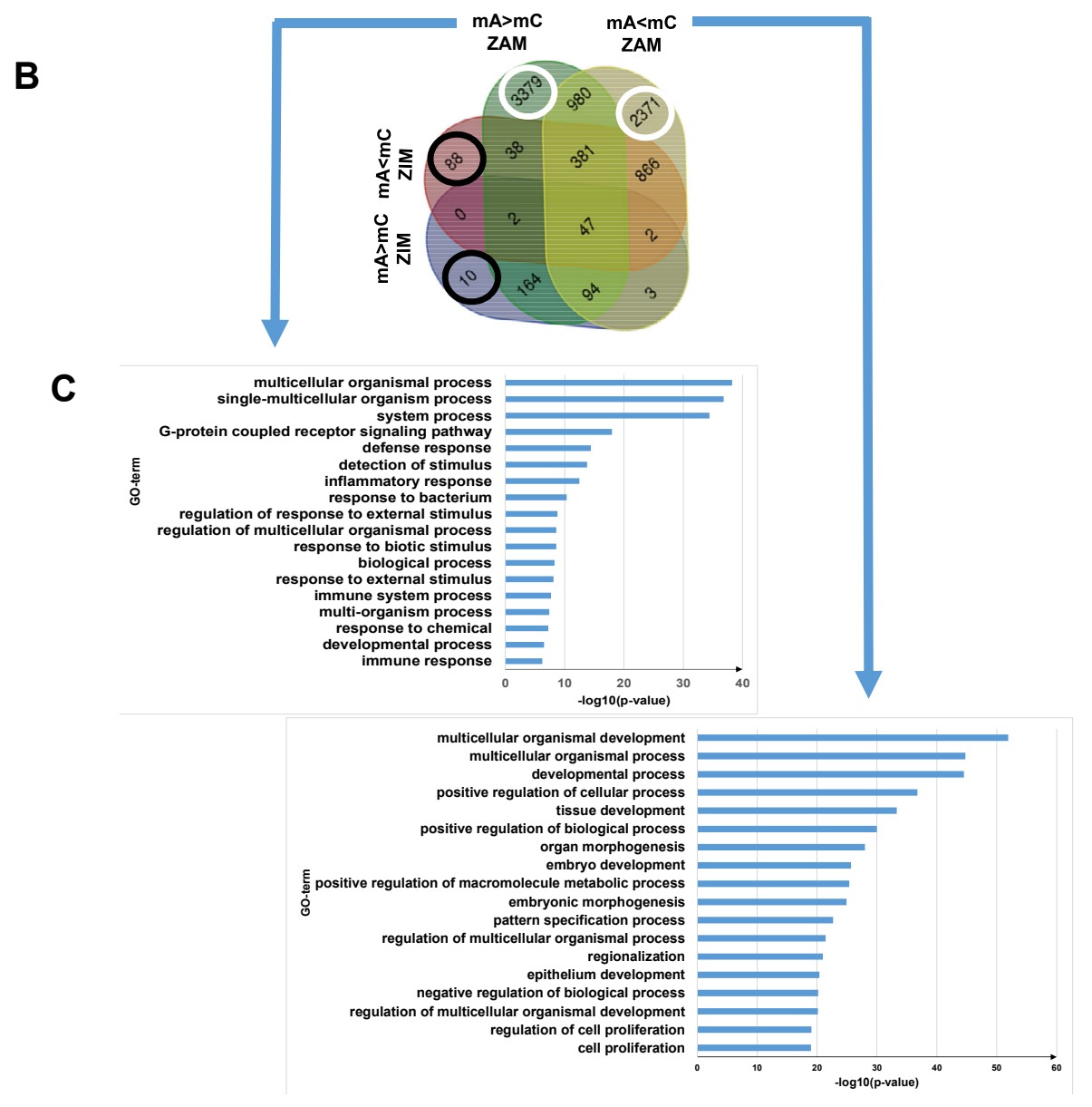

**Figure 1.** Differential cytosine methylation analysed for the zone of intact morphology (ZIM) and zone of altered morphology (ZAM). (A) Genes affected by differential methylation between the amnion and choriodecidua, separately studied for the ZIM (left) and ZAM (right). (B) Four-way Venn diagram representing the number of genes with hypermethylated cytosines in the ZIM and ZAM according to methylomic analyses. mA> mC: a specific gene was more methylated in the amnion than in the choriodecidua. mA< mC: a specific gene was less methylated in the amnion than in the choriodecidua.

*Figure 1 continued on next page*

*Figure 1 continued*

(C) Gene Ontology (GO) term classifications for genes observed specifically in the ZAM: mA> mC (left panel) and mA< mC (right panel). Bonferroni correction was conducted for p-values < 0.01.

not exhibit any statistically significant result (p-value < 0.01 with Bonferroni correction). In contrast, the specific genes modified in terms of methylation for the ZAM (total number 5750 [i.e., 3379 + 2371, as illustrated by white circles]) led to the discovery of the numerous biological processes detailed in *Figure 1C*. Specifically, the number of characterisations of an enrichment in Gene Ontology (GO) term IDs were clearly more concentrated and, thus, were more biologically significant in the case where a definite gene was more methylated in the amnion than in the choriodecidua (mA> mC). The biological process term IDs containing the most important gene number were the G protein-coupled receptor signalling pathways (such as for defence response, detection of stimulus or inflammatory response), which could be linked to sterile inflammation and the onset of parturition related to fetal membrane ruptures.

## Transcriptomic analysis of fetal membranes in the ZAM is more relevant for downregulated genes in the amnion compared with the choriodecidua

To supplement the results obtained from methylome analysis, a transcriptomic study using the same samples was performed for the ZAM to compare the differences in gene expression between the amnion and choridecidua. We observed that 501 and 145 genes specific to the ZAM were down- and upregulated, respectively, when the expression levels were compared between the amnion and choriodecidua (a log2-fold change [FC] cut-off lower or higher than 2.8 [*Figure 2*, middle panel, *Supplementary file 1c*]).

Analysis of the genes less expressed in the amnion than in the choriodecidua (i.e., log2 FC <2.8) underlined two statistically significant GO terms: *response to external stimulus* and *female pregnancy* (*Figure 2*, top panel). In the case where genes were more expressed in the amnion than in the choriodecidua (i.e., log2 FC >2.8), the biological processes exhibited no statistically significant result (p-value < 0.01 with Bonferroni correction). Analyses undertaken with uncorrected p-values did not provide relevant information because conventional tissue development or ossification was observed (*Figure 2*, bottom panel).

## Combination of transcriptomic and methylomic results in the ZAM demonstrate that genes more expressed in the choriodecidua are linked to pregnancy pathologies

By cross-analysing the results obtained from the two preceding analyses (methylome and transcriptome), a total of 26 genes were hypermethylated in the choriodecidua, and more were expressed in the amnion (*Supplementary file 1d*). Biological process (GO term) analysis was not significant, which is likely because of the small number of studied genes and because the underlying generic processes were linked only to urogenital abnormalities, not to pathological pregnancy, as confirmed by the MeSH disease terms (*Supplementary file 1e*).

Conversely, 105 genes were hypermethylated in the amnion and were more expressed in the choriodecidua (*Figure 3A* and *Supplementary file 1f*); they could be classified as MeSH disease terms linked directly to pregnancy pathologies, such as placenta diseases (trophoblastic neoplasms, preeclampsia, fetal growth retardation, or placenta accreta), female urogenital diseases, and pregnancy complications (*Supplementary file 1g*). GO term analysis clearly identified three complementary pathways: response to external stimulus, detection of LPS, and inflammatory response. Interestingly, fetal membranes were sensitive to external stimuli, such as Gram-negative bacterial molecules and LPSs, in relation to the inflammatory response. Of all the genes classified in these processes, TLR4 was the only one represented in all these biological processes and, therefore, seems to play a central role in parturition at term. To validate this in silico observation and pave the way for describing TLR4's importance, immunofluorescence experiments were first conducted to confirm the protein's presence in the amnion and choriodecidua of the ZAM (*Figure 3B*). Considering the important role of TLR4 in sterile (regarding HMGB1 release) (*Bredeson et al., 2014*) or pathological inflammation (chorioamnionitis)

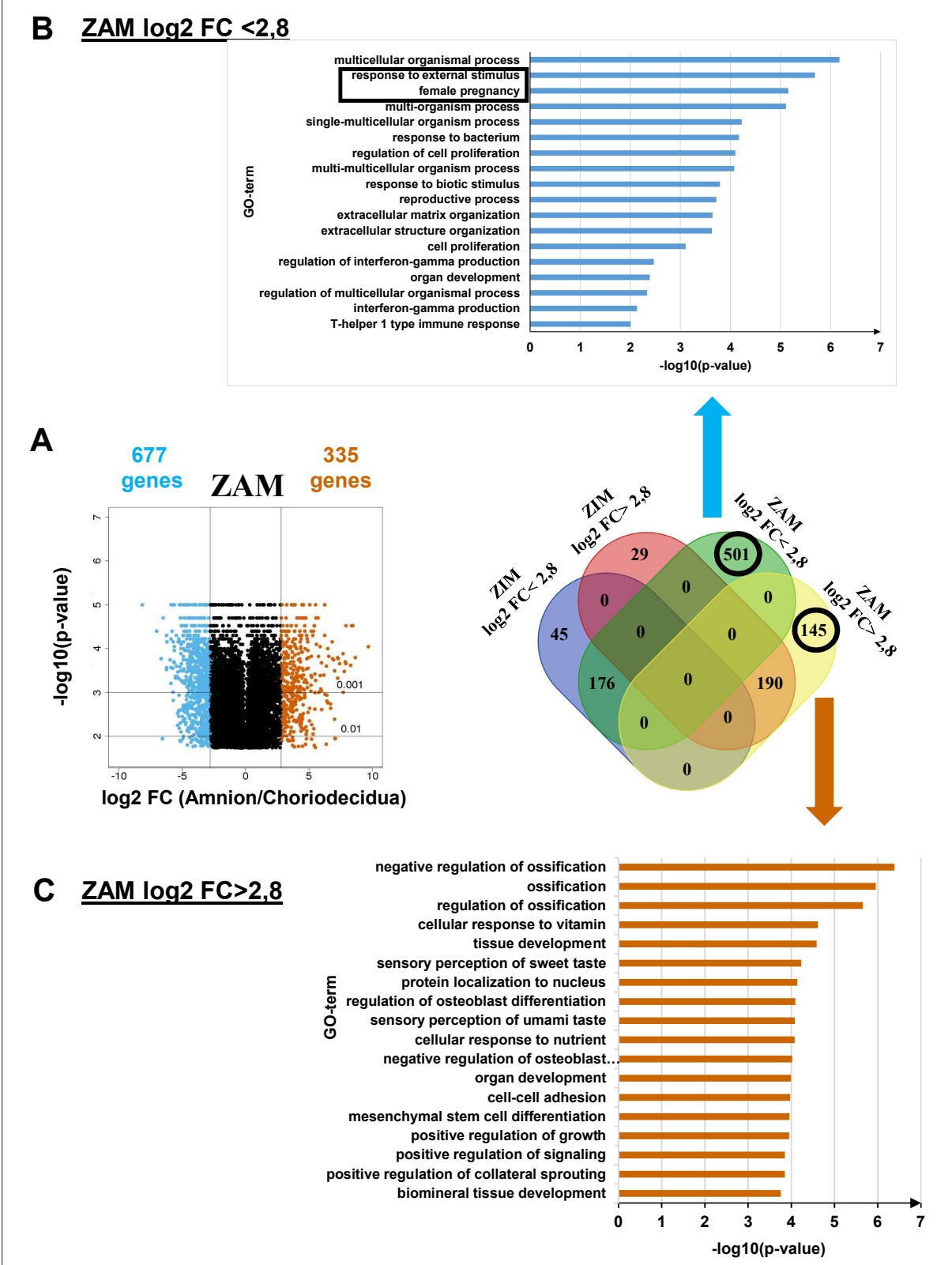

**Figure 2.** Transcriptomic assay analysed for the zone of altered morphology (ZAM) in fetal membranes. (A) Volcano plots represent the log10-adjusted p-values vs. the log2-fold change (FC). Up- and downregulated genes are shown in red and blue, respectively, limited by $|\log_2 FC| = 2.8$. They are classified in a four-way-Venn diagram representing the gene numbers in zone of intact morphology (ZIM) and ZAM analyses with $|\log_2 FC| = 2.8$. (B) Gene Ontology (GO) term classifications are shown for genes expressed only in the ZAM for log2 FC <2.8 (Bonferroni correction for p-values < 0.01). (C) GO term classifications are shown for genes expressed only in the ZAM for log2 FC >2.8 (uncorrected p-value < 0.01).

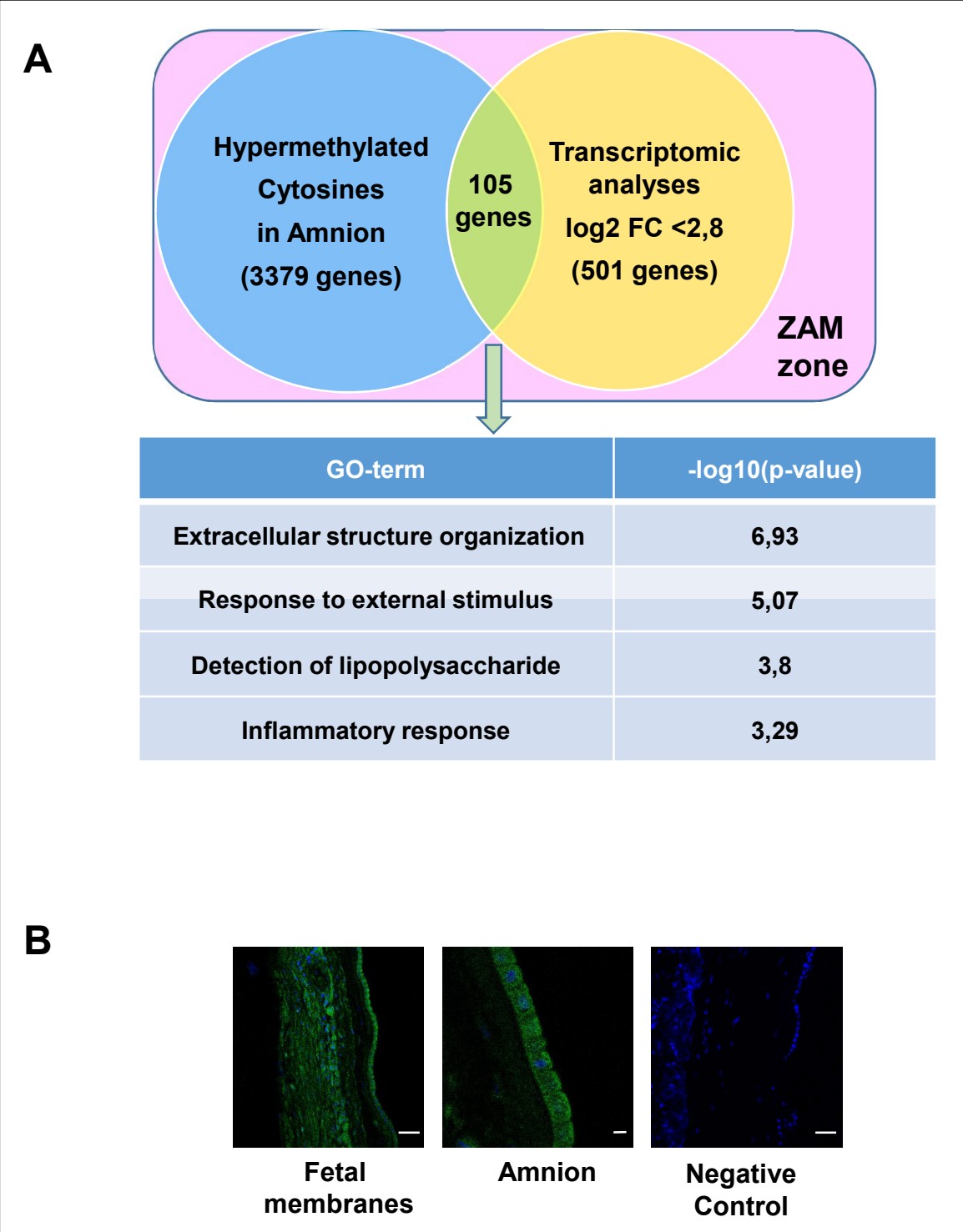

| GO-term | -log10(p-value) |
| --- | --- |
| Extracellular structure organization | 6,93 |
| Response to external stimulus | 5,07 |
| Detection of lipopolysaccharide | 3,8 |
| Inflammatory response | 3,29 |

**Figure 3.** Common genes observed between the mA> mC methylomic results and choriodecidua/amnion transcriptomic analysis in the zone of altered morphology (ZAM). (A) Gene Ontology (GO) terms' representative distribution and their log10 p-values are shown for the 105 common genes across both genome-wide studies. (B) Representative TLR4 immunofluorescence (green staining) in the ZAM of fetal membranes in confocal analyses. Cell nuclei were visualised with Hoechst (blue) staining. A negative control was used without a primary antibody. Slides were observed at ×250 magnification for total fetal membranes (left and right): scale bar: 50 µm and ×400 for the amnion (middle): scale bar: 20 µm.

The online version of this article includes the following source data for figure 3:

**Source data 1.** Raw data for toll-like receptor 4 (TLR4) immunofluorescence in the fetal membranes by confocal microscopy.

*Figure 3 continued on next page*

*Figure 3 continued*

**Source data 2.** Raw data for toll-like receptor 4 (TLR4) immunofluorescence in the amnion by confocal microscopy.

**Source data 3.** Raw data for toll-like receptor 4 (TLR4) immunofluorescence in the fetal membranes by confocal microscopy(negative control).

(*Robertson et al., 2020*), the remainder of the study focused on the characterisation and regulation of TLR4 expression in fetal membranes, as ascertained by cross-analysis of the methylomic and transcriptomic results.

## The tissue specificity of TLR4 regulation at the end of pregnancy in the weak ZAM is because of hypermethylation in the amnion

Focusing on the TLR4 genomic zone, five cytosines distributed on the promoter, body, and UTR were studied using a methylomic array. The statistical study of the cytosine methylation β-values of the nine genomic DNA samples demonstrated that the five cytosines were hypermethylated in the amnion compared with the choriodecidua (*Figure 4A*). These results were confirmed by the bisulfite treatment and enzymatic digestion (taking advantage of the Taq I restriction site present only in this CpG zone) of the amnion and choriodecidua samples, here with a focus on only one cytosine: cg 05429895 (*Figure 4B*). Furthermore, of the nine RNA samples, the link between methylation and expression revealed by the methylomic and transcriptomic arrays was confirmed by quantitative RT-PCR (qRT-PCR) (*Figure 4C*) and Western blot (*Figure 4D*) analyses, showing a higher expression of TLR4 in the choriodecidua than in the amnion (mRNA fold: 40.65 and protein fold: 4.04).

## The tissue specificity of TLR4 regulation could be because of miRNA action

Because gene expression can also be regulated post-transcriptionally by the action of mi-RNA, it would be interesting to perform in silico analysis of differentially methylated miRNA (linked to TLR4) between the amnion and choriodecidua. The results showed that two miRNA (let-7a-2 and miR-125b-1) could target the 3'UTR region of TLR4 mRNA (*Figure 5A*). Focusing on these, all the cytosines studied on the methylomic chip were situated in the 5' upstream miRNA sequence. Five of these six were statistically significant for overmethylation in the choriodecidua compared with the amnion for miR-125b-1 and for the unique one in let-7a-2 (*Figure 5B* upper and bottom panel). Using the same samples previously used for the quantification of mRNA TLR4, the expression of each pri-miRNA by qRT-PCR was checked. As expected, an overexpression of pri-miR-125b-1 was observed in the amnion compared with the choriodecidua (FC: 5.32). A weak basal expression level below the detection limit of the qPCR assay for pri-let-7a-2 did not allow for obtaining results that could reasonably be analysed (*Figure 5C*).

## Let-7a-2 and miR-125b-1 target 3'-UTR-TLR4

We first established that in the human cell line HEK293 (classically used in a gene reporter assay for testing miRNA targets), both miRNAs could decrease the amount of luciferase by targeting the 3'-UTR region of TLR4 mRNA (decrease by 40% for both miRNA tested) (*Figure 6A*). To choose a better cellular model to test this miRNA action linked with the fetal membrane environment, the relative amount of TLR4 mRNA was determined in various human amniotic cells. *Figure 6B* demonstrates that AV3 (one of the three tested amniocyte cell lines when compared with FL and WISH) was the most efficient and, therefore, was used for the experiments. For the in vitro models, AV3 (*Figure 6C*) and primary amniotic epithelial cells (*Figure 6D*) were used to confirm that both miRNAs clearly targeted TLR4 mRNA or protein accumulation, respectively. For the AV3 cell line, a significant decrease in the mRNA amount was observed from 12 hr after the transfection of let-7a-2 (decrease by 3.75) and miR-125b-1 (decrease by 10.83) or the cotransfection of let-7a-2+ miR-125b-1 (decrease by 9.5). This effect persisted only for let-7a-2 at 24 hr (decrease by 1.59), confirming the mobilisation of this cellular system. Furthermore, after 24 hr of transfection, a decrease in the amount of TLR4 protein was significantly demonstrated for let-7a-2 (decrease by 1.88); this was also the case after 48 hr for miR-125b-1 (decrease by 2.54) or for both transfections of let-7a-2+ miR-125b-1 (decrease by 2.64). The use of primary cells confirmed this physiological action with a decrease in the amount of TLR4 mRNA at 48 hr after the transfection of miR-125b-1 (decrease by 1.71) and cotransfection of let-7a-2+ miR-125b-1

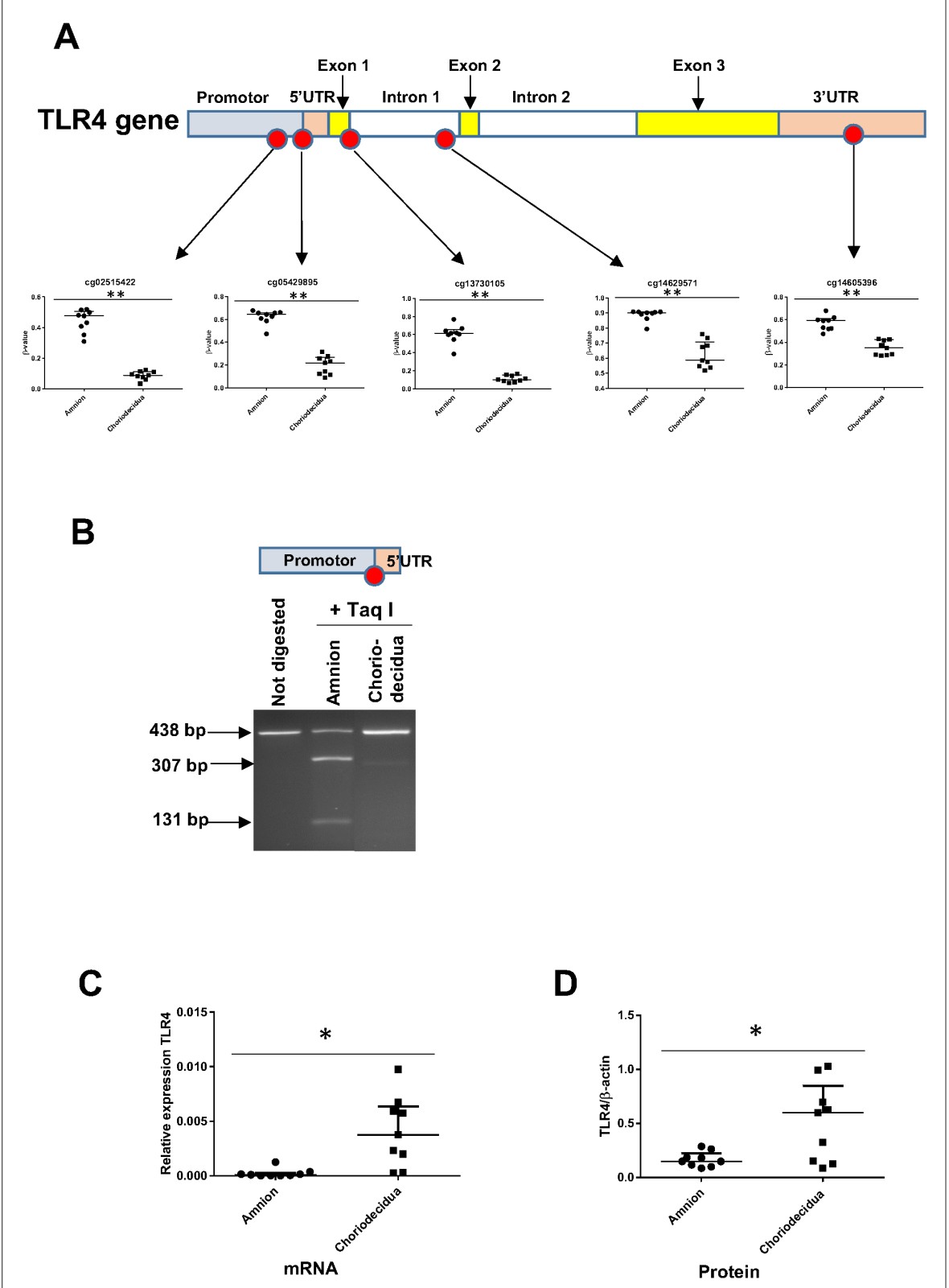

**Figure 4.** The expression level of the toll-like receptor 4 (TLR4) gene is related to its cytosine methylation level in the amnion and choriodecidua from the zone of altered morphology (ZAM). (A) The median ± interquartile ranges of the DNA methylation levels of the five cg probes on the TLR4 gene in the amnion and choriodecidua ZAMs. Each dot represents the individual β-value for one patient (n = 9). The probe set with significant differential methylation between the amnion and choriodecidua (Wilcoxon matched-pairs signed-rank test) is designated by an asterisk (**p-value < 0.01). (B)

*Figure 4 continued on next page*

*Figure 4 continued*

Control of the difference of methylation for the cg 05429895 between the amnion and choriodecidua after digestion with Taq I PCR products (438 bp) obtained after DNA bisulfite treatment. This probe, after being methylated in the amnion, was sensitive to Taq I and gave two fragments after digestion: 307 and 131 bp. (C) Relative expression of TLR4 transcripts for the nine samples of choriodecidua ZAM were significantly higher than for the amnion ZAM (Wilcoxon matched-pairs signed-rank test *p-value < 0.05). (D) The TLR4 protein was significantly overexpressed for the nine samples in the choriodecidua compared with the amnion (Wilcoxon matched-pairs signed-rank test *p-value < 0.05).

The online version of this article includes the following source data for figure 4:

**Source data 1.** Data on β-value for cg probes on the toll-like receptor 4 (TLR4) gene for *Figure 4A*.

**Source data 2.** Uncropped polyacrylamide gel for *Figure 4B*.

**Source data 3.** Quantitative RT-PCR (qRT-PCR) data and Western blot data for toll-like receptor 4 (TLR4) expression (*Figure 4C*).

(decrease by 2.16) and after 72 hr for the TLR4 protein for the three conditions (let-7a-2: decrease by 1.43; miR-125b-1: decrease by 1.71; let-7a-2 and miR-125b-1: decrease by 2.16).

## Discussion

Worldwide, preterm birth is a serious medical problem, particularly when it comes to its long-term consequences throughout the entire life of premature infants. PPROM represents one-third of pregnancies that end prematurely and principally depends on an early scale and the kinetic activation of inflammation, signalling a cascade in gestational tissues.

Because the inflammatory mechanisms of preterm and term birth are broadly similar, understanding how human fetal membranes are prepared for their physiological rupture is essential and could be a molecular mechanism abnormally exacerbated during PPROM. Studies have partially documented and established that a weak zone (ZAM) situated in the cervix zone emerges at the end of the 9-month gestation period (*McLaren et al., 1999*) as a direct consequence of global layer disorganisation and weakening. All of these phenomena are key determinants of a rupture and could be directly linked to a common denominator: the regulation and modification of gene expression levels (*Romero et al., 2006*). For the first time, a complementary high-throughput approach was applied by performing both transcriptomic and methylomic analyses of the same samples. Moreover, one strength of our work is highlighting both the global geographical (ZIM/ZAM) and tissue layer (amnion vs. choriodecidua) differences in terms of DNA methylation and gene expression.

Indeed, our methylomic study established a global profile of fetal membranes in the ZIM and ZAM samples. The number of hypermethylated genes in the amnion or choriodecidua was more important in the ZAM than in the ZIM; this allowed us to perform a GO term classification focused on the ZAM, where cell adhesion, response to a stimulus, tissue development, and reproductive processes are significantly represented. We found some similarities with transcriptomic analysis, where overexpressed genes in the choriodecidua ZAM were linked to biological adhesion (important in tissue integrity), regulation of cell proliferation, extracellular matrix organisation, and responses to internal and external stimuli.

By cross-analysing both methods for the ZAM, a link for the 105 genes overexpressed in the choriodecidua and hypomethylated in the amnion with MeSH disease terms (e.g., pregnancy complication) was revealed. Especially in GO terms, these highlighted processes could be directly linked to preparation for parturition through a sterile inflammation cascade. We then turned our attention to TLR4, a major mediator of inflammation, for the following reasons: First, this upstream gatekeeper of innate immune activation was the only gene appearing in each of the considered GO terms, as illustrated in *Figure 3A*. Second, it is an important partner of many other genes included in our list of 105 genes (see *Figure 3B*). Third, TLR4 was already known to be expressed in the fetal membranes and cervices of animals (*Gonzalez et al., 2007*; *Harju et al., 2005*; *Moço et al., 2013*) and to play a role as a key regulator in a sterile or septic inflammatory reaction in response to aggression by a pathogen-associated molecular pattern/DAMP, which leads to PPROM (*Chin et al., 2016*; *Li et al., 2010*; *Patni et al., 2007*; *Wang and Hirsch, 2003*). Fourth, TLR4 is also known to be involved in parturition in the classical term of pregnancy, activating a sterile inflammation cascade (*Choi et al., 2012*; *Wahid et al., 2015*). Fifth, it could be considered a promising therapeutic target to prevent preterm birth through better control of its proinflammatory signalling.

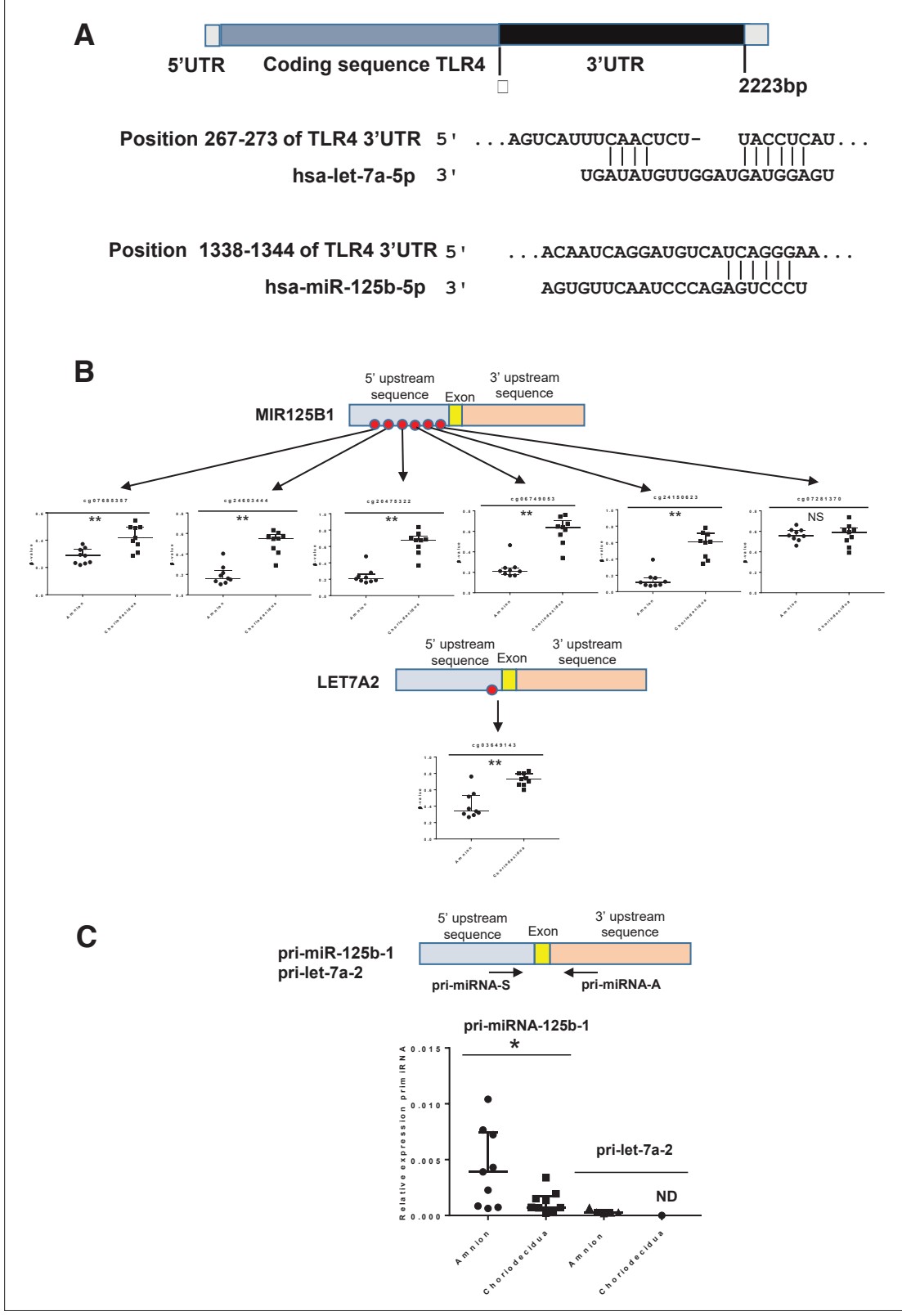

**Figure 5.** Two miRNAs potentially targeting the human 3'UTR-TLR4 (toll-like receptor 4) are differentially methylated in the zone of altered morphology (ZAM) between the amnion and choriodecidua. (A) In silico computational target prediction analysis using TargetScan of the human 3'UTR-TLR4. This zone may be targeted by gene coding for MIR125B1 and LET7A2. (B) The median ± interquartile ranges of the DNA methylation cg probe levels for the MIR125B1 (top) and LET7A2 (bottom) genes for the nine samples in the amnion and choriodecidua ZAM. Each dot represents the individual β-value

*Figure 5 continued on next page*

*Figure 5 continued*

for one patient. The probe set with significant differential methylation between the amnion and choriodecidua (Wilcoxon matched-pairs signed-rank test) is designated by an asterisk (**p-value < 0.01, NS = not significant). (C) The relative expression of pri-miR-125b-1 (n = 9) and pri-let-7a-2 (n = 5) was determined. qRT-PCR experiments performed for each zone; tissue demonstrated that pri-miR-125b-1 was significantly overexpressed in the amnion compared with the choriodecidua ZAM, as expected from the differential methylation status (Wilcoxon matched-pairs signed-rank test *p-value < 0.05). The pri-let-7a-2 amount could not be rigorously determined (ND) in the choriodecidua.

The online version of this article includes the following source data for figure 5:

**Source data 1.** Data on β-value for cg probes on the MIR125B1 and LET7A2 genes for *Figure 5B*.

**Source data 2.** Quantitative RT-PCR (qRT-PCR) data for pri-miR-125b-1 and pri-let-7a-2 (*Figure 5C*).

According to a previous study (*Krol et al., 2010*)—and as confirmed by our qPCR and Western blot quantification—TLR4 was more expressed in the choriodecidua than in the amnion (*Figure 4B and C*). Targeting the choriodecidua is interesting because it is the first breaking layer in the different steps of the rupture of membranes (*Méhats et al., 2011*). This overexpression could be because of differential methylation and/or transcription factor fixations or could be a consequence of post-transcriptional regulation by miRNAs. miRNAs play crucial regulatory roles in biological and pathological processes; they are well documented in gestational tissues, such as the placenta, endometrium, and fetal membranes (*Doridot et al., 2014*; *Gu et al., 2013*; *Kamity et al., 2019*; *Montenegro et al., 2007*; *Wang et al., 2016*), and they have begun to be considered as potential biomarkers for pregnancy pathologies (*Cretoiu et al., 2016*). Indeed, studies have already shown that uninfluenced by labour, some miRNA expressions decrease in chorioamniotic membranes with gestational age (*Montenegro et al., 2009*; *Montenegro et al., 2007*). We found two miRNAs (mir-125b-1 and let-7a-2) that were hypermethylated in the choriodecidua compared with the amnion, which could potentially target TLR4 mRNA in the amnion to decrease its expression level. They were previously unknown to be able to specifically target TLR4, leading us to demonstrate the consequences of their transfection on the quantification of TLR4 mRNA and protein on the cell line AV3 or primary amniotic epithelial cells using the luciferase 3'UTR of the TLR4 reporter gene.

Surprisingly, these two miRNAs are part of an miR-100-let7a-2 cluster host gene: MIR100HG. This cluster has never been studied in relation to the placenta or fetal membrane environment, unlike others such as C14MC, C19MC, and miR-371–3, whose expressions change during pregnancy between the whole and terminal villi (*Gu et al., 2013*; *Morales-Prieto et al., 2013*). In humans, 10 mature let-7s are synthesised and implicated in different physiological and pathological events, from embryogenesis to adult development, such as inflammatory responses and innate immunity (*Roush and Slack, 2008*). However, the latter finding could not be linked only to simple TLR4 targeting because both miRNAs were also already known to have implications for innate immunity by influencing not only the expression of interleukins and TNFα, but also cell senescence, which is another well-known phenomenon occurring at the end of fetal membrane life (*Iliopoulos et al., 2009*; *Nyholm et al., 2014*; *Schulte et al., 2011*; *Tili et al., 2007*).

On a global scale, the results obtained in the present study provide information regarding the implication of inflammation in the physiological rupture of fetal membranes. However, future studies are needed to exploit all the data accumulated during this work. Nevertheless, by focusing on a unique candidate—TLR4, which is a well-known actor in the physiological and pathological rupture of membranes—we have outlined the complex molecular process of gene regulation and have proposed a fetal membrane layer-specific model to better understand fetal membrane ruptures in the ZAM (*Figure 7*). This regulation is intriguing because it implies a double molecular mechanism of regulation (at the DNA and mRNA levels), which is apparently layer specific, though located in the same geographical area of the fetal membrane. These first results could doubtlessly be extrapolated to other genes by our high-scale studies and to gestational tissues or resident/circulating immune cells, such as neutrophils and monocytes/macrophages, which are implicated in the amplification of the inflammatory response in PPROM and physiological FM rupture (*Galaz et al., 2020*; *Gomez-Lopez et al., 2022*). Furthermore, to complete such investigations, the use of recently developed technology such as organ-on-chip will better mimic fetal–maternal communication and exchanges between tissue layer and cell types (*Richardson et al., 2020*).

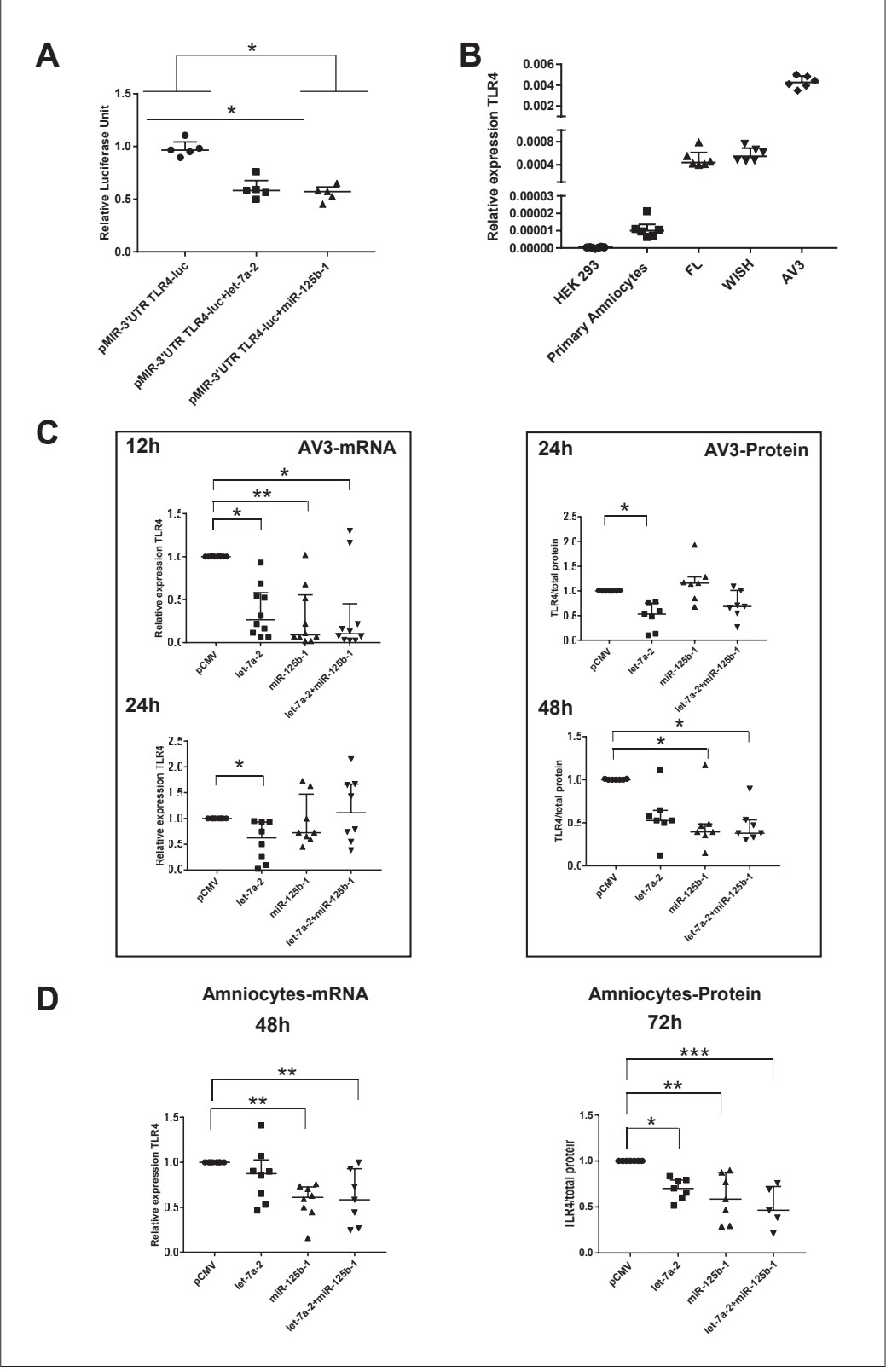

**Figure 6.** miR-125b-1 and let-7a-2 target the human 3'UTR-TLR4 (toll-like receptor 4) and decrease TLR4 expression. (A) Targeting of miR-125b-1 and let-7a-2 to the human 3'UTR of TLR4 mRNA using a Luciferase Reporter Gene Assay depending on human 3'UTR-TLR4 (pMIR-3'UTR TLR4-luc). HEK293 cells cotransfected for 48 hr with this construction and expressing plasmid of human pCMV, pre-mir125b1, or let7A2 (n = 5). Luciferase

*Figure 6 continued on next page*

*Figure 6 continued*

activity was normalised with the pRL-TK-Renilla luciferase level (median ± interquartile range, Kruskal–Wallis test with Dunn's multiple comparison test, *p-value < 0.05). The results showed that both miRNAs could target the 3'-UTR zone of the TLR4 gene and decrease luciferase quantity. (B) Determination of endogenous TLR4 expression levels in human cell lines (HEK293 from embryonic kidney and FL, WISH, and AV3 from amniocytes) and in primary amniocyte cells quantified by quantitative RT-PCR (qRT-PCR) (n = 6). (C) Effects of miR-125b-1, let-7a-2, and the combination miR-125b-1+ let-7a-2 on TLR4 mRNA expression (n = 10 for 12 hr and n = 8 for 24 hr, left) and TLR4 protein (n = 7, right) in AV3 cells. Cells were transfected with expressing plasmid of human pCMV, or pre-mir125b1, or let7A2 and miR-125b-1+ let-7a-2 for 12 and 24 hr for mRNA and 24 and 48 hr for protein (median ± interquartile range, Kruskal–Wallis test with Dunn's multiple comparison test, *p-value < 0.05, **p-value < 0.01). (D) Effects of miR-125b-1, let-7a-2, and the combination miR-125b-1+ let-7a-2 on TLR4 mRNA expression (n = 7 at least, left) at 48 hr and TLR4 protein (n = 5 at least, right) at 72 hr in primary amniocyte cells (median ± interquartile ranges, Kruskal–Wallis test with Dunn's multiple comparison test, *p-value < 0.05, **p-value < 0.01, ***p-value < 0.001).

The online version of this article includes the following source data for figure 6:

**Source data 1.** Data on luciferase activity in HEK293 cells transfected with pMIR-3'UTR toll-like receptor 4 (TLR4)-luc and pre-mir125b1 or let7A2 (*Figure 6A*).

**Source data 2.** Quantitative RT-PCR (qRT-PCR) data for toll-like receptor 4 (TLR4) expression in *Figure 6B*.

**Source data 3.** Quantitative RT-PCR (qRT-PCR) data and Western blot data for toll-like receptor 4 (TLR4) expression in AV3 cells transfected with pre-miRNA (*Figure 6C*).

**Source data 4.** Quantitative RT-PCR (qRT-PCR) data and Western blot data for toll-like receptor 4 (TLR4) expression in amniocytes transfected with pre-miRNA (*Figure 6D*).

## Materials and methods

### Human fetal membrane collection

By caesarean section, nine fetal membranes were collected at full term before labour and birth (Obstetrics Department, Estaing University Hospital, Clermont-Ferrand, France). All patients were Caucasian and presented with no pregnancy pathology (confirmed by macroscopic and microscopic

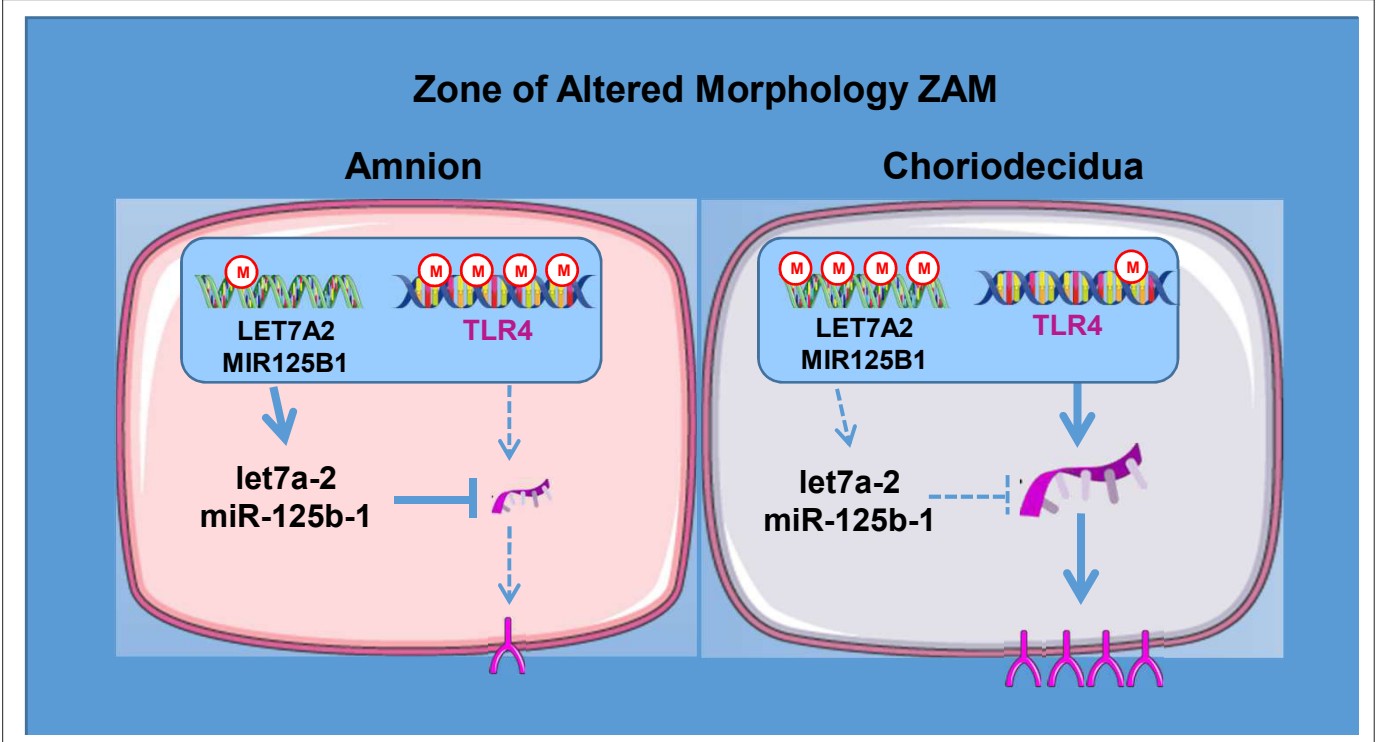

**Figure 7.** A model for the regulation of toll-like receptor 4 (TLR4) expression in the zone of altered morphology (ZAM) of human fetal membranes.

placenta analyses and histological examinations that excluded chorioamnionitis). *Supplementary file 1a* describes the patients' characteristics.

Four samples were collected from each patient (ZAM amnion, ZAM choriodecidua, ZIM amnion, and ZIM choriodecidua), as described by *Choltus et al., 2020*. Briefly, a suture sewn onto the FMs (from caesarean deliveries) in front of the cervix by the midwife allowed us to identify the ZAM; then, a 4 cm diameter circle was cut and considered the ZAM, and explants localised places away from circle boundary were considered the ZIM.

## Genome-wide DNA methylation

Total genomic DNA from the amnion and choriodecidua was extracted using the QIAampDNA Mini Kit (Qiagen, Courtaboeuf, France), here following the manufacturer's instructions. DNA concentration was determined by Qubit quantitation (Invitrogen, Thermo Fisher Scientific, Illkirch, France). Five hundred ng of extracted DNA was bisulfite treated using the EZ DNA Methylation kit (Proteigene, Saint-Marcel, France), following the manufacturer's standard protocol, before being individually hybridised using HumanMethylation450 Analysis BeadChip (Illumina, San Diego, CA), which allowed for studying the 482,421 cytosines on the human genome. This array was conducted using Helixio (Saint-Beauzire, France). Specific CpG probe methylation differences between the tissues taken from the ZIM and ZAM regions or between the amnion and choriodecidua regions were analysed. To study each cytosine, methylation differences were defined as a change in β-values (Δβ) between two conditions with p-values less than 0.05 (after applying a Student's paired t-test). The Monte Carlo method (*Metropolis and Ulam, 1949*) was then used to randomly stimulate the minimal (m) and maximal (M) Δβ-value for each chromosome. These limits permitted us to keep only the cytosine, in which Δβ was inferior to M or superior to M. Gene probes meeting the cut-off criteria for each comparison were submitted for Database for Gene Ontology (GO) enrichment (http://go.princeton.edu), in association with the REVIGO web server (http://revigo.irb.hr; as previously described by *Supek et al., 2011*) to identify the biological processes associated with the genes showing changes in methylation status.

## Human GE 8 × 60 K expression arrays

Following the manufacturer's instructions, total RNA was extracted using an RNeasy Mini Kit (Qiagen). RNA samples were quantified with a NanoDrop spectrophotometer (ThermoScientific, Thermo Fisher Scientific, Illkirch, France). RNA integrity was evaluated using a 2100 Bioanalyzer (Agilent Technologies, Les Ullis, France) and an RNA 6000 Nano Assay Kit. The mean RNA integrity number for all samples was 8.41.

A microarray (Sure Print G3 Human GE 8 × 60 K) was performed by Helixio (Saint-Beauzire, France) in accordance with Agilent Technologies. The data were analysed using Genespring GX 12.0 (Agilent Technologies).

The resulting gene lists from each pairwise comparison (ZIM vs. ZAM and amnion vs. choriodecidua, respectively) were filtered for the genes showing 2.8-fold changes at p < 0.01 using the Student's–Newman–Keuls test with Benjamini–Hochberg corrections for the false discovery rates. Raw data were analysed with R (http://R-project.org) and Bioconductor (http://www.bioconductor.org) software. A generic GO term finder (http://go.princeton.edu) and REVIGO (http://revigo.irb.hr) with up- and downregulated probes were used to analyse the biological process of GO enrichment, and GePS Genomatix software (Release 2.4.0 Genomatix Software GmbH, Munich, Germany) was used for analysis of MeSH diseases (p < 0.01).

## TLR4 immunofluorescent staining

Immunohistochemistry experiments performed on ZAM cryosections taken from the same samples (8 μm) were fixed in paraformaldehyde (4% in PBS); blocked with PBS-1X, Triton 0.1%, and SVF 10%; and incubated overnight with a monoclonal mouse anti-TLR4 antibody diluted 1:400 in PBS (Sc-293072, Santa-Cruz Biotechnology, Dallas, TX). A secondary antibody (donkey antimouse coupled with Alexa 488 Ig G [H + L]; Life Technologies, Thermo Fisher Scientific) diluted at 1:300 was incubated for 2 hr on slides. Following Hoechst nuclear staining, the samples were examined under an LSM510 Zeiss confocal microscope (Zeiss, Oberkochen, Germany). For the negative controls, sections were incubated without a primary antibody.

### Bisulfite conversion and combined restriction analysis

Total DNA from the amnion and choriodecidua explants was bisulfite treated using an EZ DNA Methylation kit (Zymo Research, Proteigene). The converted DNA samples were used as templates for the PCR using specific primers of promoter TLR4 (containing the cytosine cg 05429895; promot-F: 5' TTTAGAGAGTTATAAGGGTTATTT 3'; and promot-R: 5' CTAACA TCATCCT CACTACTTC 3'). The PCR was performed using HotStart DNA Polymerase (Qiagen), and the products were digested with Taq I (New England Biolabs, Evry, France), which could recognise CpGs. Digested products were analysed on polyacrylamide gel.

### Cell cultures

Primary amniocyte cells were collected from the amnion after trypsination and were cultured on six-well plates coated with bovine collagen type I/III (StemCell Technologies, Saint-Egrève, France), as described previously (*Marceau et al., 2006*; *Prat et al., 2015*). Human embryonic kidney (HEK) 293 cells were grown in Dulbecco's modified Eagle medium (DMEM; Gibco) supplemented with 10% fetal bovine serum (FBS, GE Healthcare), 4 mM glutamine (Gibco), 1 mM sodium pyruvate (Hyclone), 1× nonessential amino acids (Gibco), 100 units/ml of penicillin, 100 µg/ml of streptomycin, and 0.25 µg/ml of amphotericin B (Hyclone).

Human epithelial amnion cells (AV3 cells, ATCC-CCL21; FL, ATCC-CCL62; WISH, ATCC-CCL25) were cultured in DMEM/F12 (Gibco) supplemented with 10% FBS (GE Healthcare), 4 mM glutamine (Gibco), 100 units/ml of penicillin, 100 µg/ml of streptomycin, and 0.25 µg/ml of amphotericin B (Hyclone).

### Quantitative RT-PCR

Total RNA was isolated from the amnion, choriodecidua, human epithelial amnion cells (AV3, FL, and WISH) and primary amniocyte cells using an RNeasy Mini Kit (Qiagen) with DNAse I digestion, as described in the manufacturer's protocol. After quantification with a NanoDrop spectrophotometer (Thermo Fisher Scientific), cDNA was synthesised from 1 µg of RNA using a SuperScript III First-Strand Synthesis System for RT-PCR (Invitrogen, Thermo Fisher Scientific). qRT-PCRs were performed with a LightCycler 480 (Roche Diagnostics, Meylan, France) using Power SYBR Green Master Mix (Roche) and specific primers (described in *Supplementary file 1b*). Transcripts were quantified using the standard curve method. The ratio of interest (transcript/geometric) mean of two housekeeping genes (RPLP0 and RPS17) was determined. The results were obtained from at least six independent experiments, and all steps followed the MIQE guidelines (*Bustin et al., 2009*).

### Western blot

Primary amniocyte and AV3 cells were lysed in a RIPA buffer (20 mM Tris-HCl pH = 7.5, 150 mM NaCl, 1% Nonidet P40, 0.5% sodium deoxycholate, 1 mM EDTA, 0.1% SDS, and 1× Complete Protease inhibitor cocktail (Roche)) for 30 min at 4°C. The amnion and choriodecidua samples were homogenisated, as previously described (*Choltus et al., 2020*). The protein concentration of the supernatant was determined using a Pierce BCA Assay Kit (Pierce). Forty µg of denatured proteins were subjected to a Western blot analysis after 4–15% MiniPROTEAN TGX StainFree gel electrophoresis (Bio-Rad), which was followed by probing antibodies against TLR4 or β-Actin (TLR4:1:400, Sc-293072, Santa-Cruz Biotechnology, β-Actin:1:10,000, MA1-91399, Thermo Fisher Scientific). The signal was detected with a peroxidase-labelled antimouse antibody at 1:10,000 (Sc-2005, Santa-Cruz Biotechnology) and visualised with ECL or ECL2 Western blotting substrate (Pierce) using a ChemiDoc MP Imaging System and Image Lab software (Bio-Rad). The results were obtained from at least seven independent experiments, and the relative TLR4 ratio was expressed as a function of β-Actin or total protein loaded by well.

### 3'UTR-hTLR4 pMIR REPORT luciferase plasmid

The 3'UTR hTLR4 (2,223 bp, NM 138554) was amplified from 100 ng of genomic amnion DNA with 3'UTR-TLR4 primers containing Spe I restriction sites to facilitate subcloning: forward GAGAACTA GTAGAGGAAAAATAAAAACCTCCTG and reverse GAGA ACTAGTTTGATATTATAAAACTGCATAT ATTTA. PCR was performed with Phusion high-fidelity DNA polymerase (New England Biolabs) according to the manufacturer's instructions. After purification and digestion with Spe I (New England

Biolabs), the fragment was subcloned into the Spe I site of the pMIR REPORT luciferase vector (Ambion, Thermo Fisher Scientific) to generate the construct pMIR-3'UTR-hTLR4. The insert sequence was checked by sequencing (Eurofins Genomics, Ebersberg, Germany).

### In silico miRNA analysis

Putative miRNAs targeting the 3'UTR-TLR4 target gene miRNA were screened using the public database miRWalk with TargetScan, RNA22, and miRanda.

### Dual-Luciferase Reporter Assays

HEK293 cells were cultured in six-well plates and transiently transfected at an 80–90% confluence using a Lipofectamine 3000 Transfection Kit (Invitrogen, Thermo Fisher Scientific). Each well received 100 ng of the pMIR REPORT Luciferase vector containing the 3'UTR-hTLR4 (without a negative control) in combination with 1 µg pCMV-MIR or pCMV-pre-mir125b1 (OriGene Technologies, Rockville, MD) or let7A2 generated in the laboratory and with 50 ng of pRL-TK Renilla luciferase plasmid (Promega). Briefly, the pre-let7A2 was amplified from 100 ng of genomic amnion DNA with pre-let7A2 primers containing SgfI/MluI restriction sites to facilitate subcloning: forward GAGAGCGA TCGCTCGTCAACAGATATCAGAAGGC and reverse GAGAACGCGTAATGCTGCATTTTTTGTG ACAATTT. PCR was performed with Advantage-HD DNA polymerase (Takara Bio, Saint-Germain-en-Laye, France), according to the manufacturer's instructions. After purification and digestion with SgfI/MluI (New England Biolabs), the fragment was subcloned into the SgfI/MluI site of the pCMV-MIR vector to generate the construct pCMV-pre-let7A2. The insert sequence was checked by sequencing (Eurofins Genomics, Ebersberg, Germany). Luciferase activity was measured using the Dual-Luciferase Reporter Assay System (Promega, Charbonnières-les-Bains, France) 48 hr after transfection according to the manufacturer's instructions using a Sirius luminometer (Berthold, Thoiry, France). Transfection efficiencies were normalised to the Renilla luciferase activities in the corresponding wells and reported to the control condition. Data were extracted from at least three experiments, each of which was performed in triplicate.

### Transfection of primary amniocytes and AV3 cells

Primary amniocytes and AV3 were cultured in six-well plates. At 80–90% confluence, the cells were transiently transfected using a Lipofectamine 3000 Transfection Kit (Invitrogen, Thermo Fisher Scientific) according to the protocol with 1 µg expressing plasmid of human pCMV-pre-mir125b1, pre-let7A2, or a control (pCMV-MIR) purchased from OriGene. After 48 or 72 hr of culturing, the total RNA and proteins were collected and used for the qRT-PCR or Western blot experiments, respectively. Data were extracted from five experiments.

### Statistical analysis

The results were analysed using PRISM software (GraphPad Software Inc, San Diego, CA). The quantitative data were presented as the medians ± interquartile ranges according to a Shapiro–Wilks test. Non-normally distributed data between the two groups were studied with a Wilcoxon matched-pairs signed-rank test for paired samples. To study several independent groups, a Kruskal–Wallis test was performed, followed by Dunn's multiple comparison test. Values were considered significantly different at $p < 0.05$ (*), $p < 0.01$ (**) or $p < 0.001$ (***) throughout.

## Acknowledgements

LB was supported in carrying out this work by the Auvergne-Rhone-Alpes region's Jeune Chercheur framework. This research was conducted with the scientific support and expertise of Helixio (Saint-Beauzire, France).

The authors thank Scribendi proofreading services (Canada) for their English language editing of the initial and revised version of the article.

## Additional information

### Funding

| Funder | Grant reference number | Author |
|---|---|---|
| Auvergne Rhone-Alpes Region Jeune Chercheur Framework | Not applicable | Loïc Blanchon |

The funders had no role in study design, data collection and interpretation, or the decision to submit the work for publication.

### Author contributions

Corinne Belville, Conceptualization, Investigation, Methodology, Writing - original draft; Flora Ponelle-Chachuat, Formal analysis, Methodology, Software; Marion Rouzaire, Christelle Gross, Denis Gallot, Investigation, Methodology; Bruno Pereira, Conceptualization, Methodology, Software; Vincent Sapin, Methodology, Supervision, Writing - original draft; Loïc Blanchon, Funding acquisition, Methodology, Project administration, Supervision, Writing - original draft

### Author ORCIDs

Loïc Blanchon (iD) http://orcid.org/0000-0001-8842-0162

### Ethics

Ethics statementsThis study was approved by the "Comité de Protection des Personnes (CPP)" under the reference: AU-765 and Codecoh number: DC-2008–558. Informed consent was obtained from the participants. The experiments conformed to the principles set in the World Medical Association Declaration of Helsinki.

### Decision letter and Author response

Decision letter https://doi.org/10.7554/eLife.71521.sa1
Author response https://doi.org/10.7554/eLife.71521.sa2

## Additional files

### Supplementary files
• Supplementary file 1. Supplementary tables 1a to g.

• Transparent reporting form

### Data availability

- Term W/O labor Fetal membrane methylation status results are available at Arrayexpress :(http://www.ebi.ac.uk/arrayexpress/experiments/E-MTAB-10520). - Term W/O labor Fetal membrane mRNA expression results are available at Arrayexpress : (http://www.ebi.ac.uk/arrayexpress/experiments/E-MTAB-10516).

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
