## [Decision Letter]

**Decision letter after peer review:**

Thank you for submitting your article "TLR4 regulation in human fetal membranes as an explicative mechanism of a pathological preterm case" for consideration by *eLife*. Your article has been reviewed by 3 peer reviewers, and the evaluation has been overseen by Y M Dennis Lo as the Senior Editor and Reviewing Editor. The reviewers have opted to remain anonymous.

Essential revisions:

1. The study is highly descriptive and the authors should consider that sterile inflammation and microbe-associated inflammation induce distinct transcriptomic profiles in human fetal membranes, which was recently reported in Infect Immun. 2021 Apr 16;89(5):e00819-20. The authors could consider utilizing these publicly available data to enhance the translational value of their hypothesis.

2. There are currently available animal models of premature rupture of membranes (e.g., ultrasound-guided or mini-laparotomy). The authors should consider adding in vivo experiments to prove causality between TLR4 and rupture of membranes.

3. The authors focus on the rupture zone of the membranes, but fail to explain how the regulation of TLR4 mediates the process of membrane rupture.

4. The amnion layer is structurally different from the chorion. The chorion is infiltrated by innate immune cells during sterile and microbe-associated inflammation. The authors should consider evaluating the process of rupture of membranes in a more multi-dimensional manner, in which amnion, chorion, and immune cells are implicated.

5. The abstract, introduction and discussion would benefit from some rewriting. This would align the phraseology with English speakers, making some areas of these specific sections easier to follow.

6. Some of the text has a different color, or is highlighted and this need to be changed.

7. The data in figure 1 alludes to hypermethylated genes, using the approach is it possible to also indicate how many genes are hypomethylated? Line 135 in the results hints that this was analyzed this too.

8. Both in figures 1 and 2 many genes are shared by the ZIM and ZAM. Although these genes may not speak to the region specificity and what may be differentially occurring in the ZAM, they may inform about changes in the tissue at this gestational age globally in the tissues, will these be indicated in the supplementary data?

9. Written numerical data would improve the ability to follow the data in the results narrative and set order of magnitude context.

10. The final paragraph of the results describes the use of primary cells to confirm the decrease in TLR4 (line 236 onward) but the data is not shown. More description of the details would be helpful.

11. The data indicating the importance of TLR4 is intriguing, as is the data showing that the lower (or hypomethylated) TLR4 and MIR forms are more functional. This is summarized in figure 7 but not really discussed in the final section of the paper. This should be rectified.

12. The way in which the ZAM and ZIM regions are confirmed is not described in the methods. The reference describes the tissue collection. Were they marked or dyed in some way?

13. Abstract – Should include an objective section to the rationale for this study and how it was done.

14. One major concern is that how reliable is an examination of the transcriptome and methylome of a membrane (regardless of zone) that are from term and not ruptured. How can this be related to pathologies? Isn't it better to create an in vitro model of disease state and then examine using explants? How do authors control cervical immune cell infiltration and their impact on ZAM?

15. Introduction is way too long and misses several recent references on PPROM pathology. This section needs to be trimmed by 2/3rds and update references and provide a hypothesis and objectives.

16. Several sections are unnecessary and not accurate.

E.g. Page 5 – line117 – TLR4 is not just for *E. coli* but for Gram negative bacteria and *E. coli* infection is very rare in PPROM. Several such misleading concepts and dogmatic statements are included. Please edit them out.

Page 8 – line 156 – …..more relevant and specific – please expand and explain this better.

TLR-4 and LPS – it is not clear why authors used LPS. TLR4 is a receptor for several DAMPs and other molecules and even certain viral particles. This should be discussed and generation of DAMPs in sterile environment should be included.

Page 11; line 229 – this is not a physiologic model. Please delete.

Page 11 – lines 233 23 – it is unclear what is meant by quick action and how is this measured without a functional assay? Change in transcript alone does not necessarily indicate a functional impact.

Page 14 – line 310 – please delete 'causal effect'. This study did not examine any causal effect. Mere association and the model tested with LPS is not totally relevant in PPROM.

17. Please delete the last paragraph in discussion. A therapy to block TLR4 is unlikely to have any impact on PPROM. PPROM results from underlying cellular or matrix derangements and is not necessarily caused by infection as a primary cause. It is a localized inflammation than systemic inflammation.

*Reviewer #1 (Recommendations for the authors):*

1. The study is highly descriptive and the authors should consider that sterile inflammation and microbe-associated inflammation induce distinct transcriptomic profiles in human fetal membranes, which was recently reported in Infect Immun. 2021 Apr 16;89(5):e00819-20. The authors could consider utilizing these publicly available data to enhance the translational value of their hypothesis.

2. There are currently available animal models of premature rupture of membranes (e.g., ultrasound-guided or mini-laparotomy). The authors should consider adding in vivo experiments to prove causality between TLR4 and rupture of membranes.

3. The authors focus on the rupture zone of the membranes, but fail to explain how the regulation of TLR4 mediates the process of membrane rupture.

4. The amnion layer is structurally different from the chorion. The chorion is infiltrated by innate immune cells during sterile and microbe-associated inflammation. The authors should consider evaluating the process of rupture of membranes in a more multi-dimensional manner, in which amnion, chorion, and immune cells are implicated.

*Reviewer #2 (Recommendations for the authors):*

The abstract, introduction and discussion would benefit from some rewriting. This would align the phraseology with English speakers, making some areas of these specific sections easier to follow.

Some of the text has a different color, or is highlighted and this need to be changed.

The data in figure 1 alludes to hypermethylated genes, using the approach is it possible to also indicate how many genes are hypomethylated? Line 135 in the results hints that this was analyzed this too.

Both in figures 1 and 2 many genes where shared by the ZIM and ZAM. Although these genes may not speak to the region specificity and what may be differentially occurring in the ZAM, they may inform about changes in the tissue at this gestational age globally in the tissues, will these be indicated in the supplementary data?

Written numerical data would improve the ability to follow the data in the results narrative and set order of magnitude context.

The final paragraph of the results describes the use of primary cells to confirm the decrease in TLR4 (line 236 onward) but the data is not shown. If this is the final decision, is may simply be indicated in parenthesis.

The data indicating the importance of TLR4 is intriguing, as is the data showing that the lower (or hypomethylated) TLR4 and MIR forms are more functional. This is summarized in figure 7 but not really discussed in the final section of the paper.

The way in which the ZAM and ZIM regions are confirmed is not described in the methods. The reference describes the tissue collection. Where they marked or dyed in some way?

*Reviewer #3 (Recommendations for the authors):*

Abstract – Should include an objective section to the rationale for this study and how it was done.

My major concern is that how reliable is a examining transcriptome and methylome of a membrane (regardless of zone) that are from term and not ruptured. How can this be related to pathologies? Isn't it better to create an in vitro model of disease state and then examine using explants? How do authors control cervical immune cell infiltration and their impact on ZAM?

Introduction is way too long and misses several recent references on PPROM pathology. This section needs to be trimmed by 2/3rds and update references and provide a hypothesis and objectives.

Several sections are unnecessary and not accurate.

E.g. Page 5 – line117 – TLR4 is not just for *E. coli* but for Gram negative bacteria and *E. coli* infection is very rare in PPROM. Several such misleading concepts and dogmatic statements are included. Please edit them out.

Page 8 – line 156 – …..more relevant and specific – please expand and explain this better.

TLR-4 and LPS – it is not clear why authors used LPS. TLR4 is a receptor for several DAMPs and other molecules and even certain viral particles. This should be discussed and generation of DAMPs in sterile environment should be included.

Page 11; line 229 – this is not a physiologic model. please delete.

page 11 – lines 233 23 – it is unclear what is meant by quick action and how is this measured without a functional assay? Change in transcript alone does not necessarily indicate a functional impact.

Page 14 – line 310 – please delete 'causal effect'. This study did not examine any causal effect. Mere association and the model tested with LPS is not totally relevant in PPROM.

Please delete the last paragraph in discussion. A therapy to block TLR4 is unlikely to have any impact on PPROM. PPROM results from underlying cellular or matrix derangements and not necessarily infection (as primary cause). It is a localized inflammation than systemic inflammation.

The manuscript is very hard to read and the writing needs to be improved. The manuscript needs better organization and lack cohesiveness.

Reference list needs to be updated substantially and should include recent work on PPROM.

---

## [Author Response]

Reviewer #1 (Recommendations for the authors):1. The study is highly descriptive and the authors should consider that sterile inflammation and microbe-associated inflammation induce distinct transcriptomic profiles in human fetal membranes, which was recently reported in Infect Immun. 2021 Apr 16;89(5):e00819-20. The authors could consider utilizing these publicly available data to enhance the translational value of their hypothesis.

We agree and decided to add the proposed article and modify the sentence of the first submitted version to take this suggestion into account (line 306-307).

2. There are currently available animal models of premature rupture of membranes (e.g., ultrasound-guided or mini-laparotomy). The authors should consider adding in vivo experiments to prove causality between TLR4 and rupture of membranes.

We agree that animal models are available. Nevertheless, we considered that such models are far from the human reality. In fact, animal models are often used for fetal membrane studies, but they are different regarding pregnancy physiology, structure and uterine environment, which hamper their use. In addition, these animal models are usually based on the injection of bacterial products. However, it is well known that the percentage of PPROM associated with microbial invasion can vary based on the weeks of gestation. In fact, early gestational ages are clearly linked to high microbial associated intra-amniotic inflammation prevalence (64.3% when <25 WGA), whereas this percentage subsequently decreases throughout gestation (Romero et al., 2015), reaching one-third at term, which is better linked with the gestational stage of their study. Furthermore, the quantification of TLR4 mRNA expression and protein in the case of PPROM without chorioamnionitis compared with term no labor without chorioamnionitis has already been carried out (Kim et al., 2004), indicating an absence of a clear link between chorioamnionitis and TLR4 expression. These observations support the fact that the TLR4 model in physiological rupture could be transposed—at least in part—to sterile PPROM and initiated by the presence of alarmins (*i.e.,* HMGB1) and their binding to such type of receptors. Indeed, TLR4 is now well described as being stimulated by ligands other than LPS, such as HMGB1, a member of the DAMPs (Robertson et al., 2020). Finally, in an animal model of PPROM, an article underlined the importance of TLR4 in preterm labor by using TLR4 mice mutants in a sterile context (Wahid et al., 2015). This is why we decided (i) to not consider the proposal of the reviewer concerning animal models and (ii) removed Figure 3C (expression of TLR4 in the presence of LPS from bacterial origin), which helped keep a greater focus on the physiological rupture of fetal membranes without the involvement of bacterial presence.

3. The authors focus on the rupture zone of the membranes, but fail to explain how the regulation of TLR4 mediates the process of membrane rupture.

We agree with your comment; however, ‘how the regulation of TLR4 mediates the process of membrane rupture is not the topic of the manuscript. In addition, this has already been well established in previous publications. Nevertheless, we added a sentence in the introduction part between the lines lines 97-100: ‘The mechanisms implying TLR4 in the physiological or pathological rupture of membrane in case of PPROM are well known. Triggering TLR4 will lead to NF-κB activation, leading to an increase of the release of proinflammatory cytokine, concentration of matrix metalloprotease and prostaglandin, which are well established actors of fetal membrane rupture (Robertson et al., 2020).

4. The amnion layer is structurally different from the chorion. The chorion is infiltrated by innate immune cells during sterile and microbe-associated inflammation. The authors should consider evaluating the process of rupture of membranes in a more multi-dimensional manner, in which amnion, chorion, and immune cells are implicated.

According to the reviewer’s recommendation, at the end of the discussion, we added a sentence considering the essential fetal–maternal dialogue between amnion and choriodecidua, tissue layers and cell diversity: ‘Furthermore, to complete such investigations, the use of recently developed technology such as organ-on-chip (OOC) will better mimic fetal–maternal communication and exchanges between tissue layer and cell types’ (see lines 307-309).

Reviewer #2 (Recommendations for the authors):The abstract, introduction and discussion would benefit from some rewriting. This would align the phraseology with English speakers, making some areas of these specific sections easier to follow.Some of the text has a different color, or is highlighted and this need to be changed.

The introduction and discussion are modified, as requested. For English language editing, the paper was submitted to Scribendi for Editing and Proofreading (www.scribendi.com). This service is commonly used by the team to improve English articles prepared by the research team before final submission; this has been added in the acknowledgement part of the manuscript. The highlighted portions and different colours of the text are now suppressed.

The data in figure 1 alludes to hypermethylated genes, using the approach is it possible to also indicate how many genes are hypomethylated? Line 135 in the results hints that this was analyzed this too.

We thank the reviewer for this relevant comment. In fact, Figure 1B (Venn diagram) focused on the ZAM zone because there is no significant changes in methylation status in the ZIM zone. Concerning the ZAM, in Figure 1B, the left-upper part indicates genes where the methylation status is higher in amnion compared with the choriodecidua (*i.e.*, 3379). For the right-upper part, genes are hypermethylated in the choriodecidua compared with the amnion (*i.e.*, 2371). In the middle part, the 980 genes encountered in ZAM zone do not show a methylation difference between these two layers. To be exhaustive, we made three lists of these genes. To future readers, we have provided a supplementary file 2 containing a sheet regarding methylome ZAM (three lists of genes) (sheet 1).

Both in figures 1 and 2 many genes where shared by the ZIM and ZAM. Although these genes may not speak to the region specificity and what may be differentially occurring in the ZAM, they may inform about changes in the tissue at this gestational age globally in the tissues, will these be indicated in the supplementary data?

Thank you for this comment. We decided to provide a supplementary file 2 with a sheet regarding transcriptome (two lists of common genes). It details the genes that are commonly under- or overexpressed in the ZIM and ZAM when comparing the amnion to choriodecidua in function of fold change (log2 FC). We did the same for genes overexpressed in the choriodecidua (sheet 2).

Written numerical data would improve the ability to follow the data in the results narrative and set order of magnitude context.

We agree with this comment and added numerical data of the median +/- interquartile range in the text, when justified (Figure 4C and 4D, Figure 5C, Figure 6A, C and D).

The final paragraph of the results describes the use of primary cells to confirm the decrease in TLR4 (line 236 onward) but the data is not shown. If this is the final decision, is may simply be indicated in parenthesis.

If we understand this recommendation, the requested results concerning the primary cells were already detailed in Figure 6D and described at the end of correponding paragraph. To be more relevant, we separated the text of Figure 6C (AV3 cell line) and Figure 6D (primary amniotic cells).

The data indicating the importance of TLR4 is intriguing, as is the data showing that the lower (or hypomethylated) TLR4 and MIR forms are more functional. This is summarized in figure 7 but not really discussed in the final section of the paper.

We thank the reviewer for this comment. To complete Figure 7, at the end of the discussion, we added the following paragraph: ‘This regulation is intriguing because it implies a double molecular mechanism of regulation (at the DNA and mRNA levels), which is apparently layer specific, though located in the same geographical area of the fetal membrane’ (line 300-302). The latter could doubtlessly be extrapolated to other genes by our high-scale studies and to gestational tissues or resident/circulating immune cells, such as neutrophils, which are implicated in the amplification of the inflammatory response in late gestation’. Furthermore, we specified this more in the figure and the legend for the model concerning the ‘ZAM zone’.

The way in which the ZAM and ZIM regions are confirmed is not described in the methods. The reference describes the tissue collection. Where they marked or dyed in some way?

To complete the methods used to differentiate and collect the ZIM and ZAM zones, we decided to add some more information at the end of the paragraph entitled ‘Human fetal membrane collection’ in the Materials and methods part. The text added is as follows: ‘Briefly, a suture sewn onto the FMs (from caesarean deliveries) in front of the cervix by the midwife allowed us to identify the ZAM; then, a 4 cm diameter circle was cut and considered the ZAM, and explants localised places away from circle boundary were considered the ZIM’. This method has been commonly used, validated and published by the team for several years.

Reviewer #3 (Recommendations for the authors):Abstract – Should include an objective section to the rationale for this study and how it was done.

The ‘instructions for authors’ never mentions that the abstract should contained subheadings (only for medical publications) or specific parts such as ‘objective and rationale’. Furthermore, we believe that an explanation of how the study is conducted is already present in the abstract, so we decided to not include the modification proposed by the reviewer.

My major concern is that how reliable is a examining transcriptome and methylome of a membrane (regardless of zone) that are from term and not ruptured. How can this be related to pathologies? Isn't it better to create an in vitro model of disease state and then examine using explants? How do authors control cervical immune cell infiltration and their impact on ZAM?

Following these comments, we would like to bring up the following:

– We aim to provide to the scientific community an exhaustive list of genes that could be differentially regulated in term of transcription between the ZAM and ZIM in both FM layers.

– TLR4 was underlined and is already known as an actor in FM weakening (Robertson et al., 2020). This reference has been added in the new version of the article.

– TLR4 could be activated in sterile or microbial inflammation. In fact, early gestational ages are clearly linked to high-microbial-associated intra-amniotic inflammation prevalence (64.3% when <25 WGA), whereas this percentage subsequently decreases throughout gestation (Romero et al., 2015), reaching to one-third at term, which is more in line with the gestational stage of the current study. These observations support the fact that the TLR4 model in physiological rupture could be transposed—at least in part—to PPROM.

– Concerning cervical immune cell infiltration and their impact on the ZAM, we considered that the present article is not directly linked to this because we compare globally the layer and zone that contain a ‘pool’ of different cell types, such as immune cells. Such information regarding immune response were already published and investigated (Marcellin et al., 2017).

Introduction is way too long and misses several recent references on PPROM pathology. This section needs to be trimmed by 2/3rds and update references and provide a hypothesis and objectives.

Based on this comment, the introduction is now shorter than the initial version. Furthermore, newer references have been added in this version.

Several sections are unnecessary and not accurate.E.g. Page 5 – line117 – TLR4 is not just for *E. coli* but for Gram negative bacteria and *E. coli* infection is very rare in PPROM. Several such misleading concepts and dogmatic statements are included. Please edit them out.

We agree with this comment. *E. coli* was replaced in the introduction with ‘Gram-negative bacteria’. Furthermore, a new recent reference was added in the text regarding the implication of TLR4 in the case of preterm birth and fetal inflammatory injury (Robertson et al., 2020).

Page 8 – line 156 – …..more relevant and specific – please expand and explain this better.

We agree with the comment and modified the sentence to be clearer while quickly focusing on the significant results.

TLR-4 and LPS – it is not clear why authors used LPS. TLR4 is a receptor for several DAMPs and other molecules and even certain viral particles. This should be discussed and generation of DAMPs in sterile environment should be included.

We agree with this comment. Our choice to present this figure was only to confirm that TLR4, when activated by LPS, leads to a proinflammation activation in amnion and choriodecidua. Hence, we decided to remove Figure 3C in the new version of the manuscript. Furthermore, we added two references at the end of this part to better justify the choice to focus on TLR4 global transcriptional regulation.

Page 11; line 229 – this is not a physiologic model. please delete.

This has been carried out in the new version.

page 11 – lines 233 23 – it is unclear what is meant by quick action and how is this measured without a functional assay?

We agree with the comment. We replaced ‘quick action’ with ‘mobilisation’ to be more general and avoid this ‘speed notion’.

Change in transcript alone does not necessarily indicate a functional impact.Page 14 – line 310 – please delete 'causal effect'. This study did not examine any causal effect. Mere association and the model tested with LPS is not totally relevant in PPROM.

We thank the reviewer for this comment. Nevertheless, this part of the sentence is not the ‘causal effect’ but ‘causal information’, but we decided to delete ‘causal’ and keep only ‘information’.

Please delete the last paragraph in discussion. A therapy to block TLR4 is unlikely to have any impact on PPROM. PPROM results from underlying cellular or matrix derangements and not necessarily infection (as primary cause). It is a localized inflammation than systemic inflammation.

We agree with the comment and deleted the end of the discussion part.

The manuscript is very hard to read and the writing needs to be improved. The manuscript needs better organization and lack cohesiveness.

We thank the reviewer for this comment. First, we simplified the first submitted version of the manuscript (particularly the introduction part). Second, we know that sometimes, high-scale studies could be complex to interpret when there are a lot of data available. This why we decided to limit the number of figures and tables and to offer in a public manner all the obtained data. Third, the paper was submitted to Scribendi for Editing and Proofreading (www.scribendi.com). This service is commonly used by the team to improve English articles prepared by the research team before final submission; this has been added in the acknowledgement part of the manuscript.

Reference list needs to be updated substantially and should include recent work on PPROM.

This was done along with the addition of recent and supplementary references in the new version of the manuscript.

References:

Bredeson S, Papaconstantinou J, Deford JH, Kechichian T, Syed TA, Saade GR, Menon R. 2014. HMGB1 promotes a p38MAPK associated non-infectious inflammatory response pathway in human fetal membranes. *PLoS One* 9:e113799. doi:10.1371/journal.pone.0113799

Gomez-Lopez N, Romero R, Galaz J, Bhatti G, Done B, Miller D, Ghita C, Motomura K, Farias-Jofre M, Jung E, Pique-Regi R, Hassan SS, Chaiworapongsa T, Tarca AL. 2021. Transcriptome changes in maternal peripheral blood during term parturition mimic perturbations preceding spontaneous Preterm birth†. *Biol Reprod* ioab197. doi:10.1093/biolre/ioab197

Kim YM, Romero R, Chaiworapongsa T, Kim GJ, Kim MR, Kuivaniemi H, Tromp G, Espinoza J, Bujold E, Abrahams VM, Mor G. 2004. Toll-like receptor-2 and -4 in the chorioamniotic membranes in spontaneous labor at term and in preterm parturition that are associated with chorioamnionitis. *American Journal of Obstetrics and Gynecology* 191:1346–1355. doi:10.1016/j.ajog.2004.07.009

Marcellin L, Schmitz T, Messaoudene M, Chader D, Parizot C, Jacques S, Delaire J, Gogusev J, Schmitt A, Lesaffre C, Breuiller-Fouché M, Caignard A, Vaiman D, Goffinet F, Cabrol D, Gorochov G, Méhats C. 2017. Immune Modifications in Fetal Membranes Overlying the Cervix Precede Parturition in Humans. *JI* 198:1345–1356. doi:10.4049/jimmunol.1601482

Robertson SA, Hutchinson MR, Rice KC, Chin P-Y, Moldenhauer LM, Stark MJ, Olson DM, Keelan JA. 2020. Targeting Toll-like receptor-4 to tackle preterm birth and fetal inflammatory injury. *Clin Transl Immunology* 9:e1121. doi:10.1002/cti2.1121

Romero R, Miranda J, Chaemsaithong P, Chaiworapongsa T, Kusanovic JP, Dong Z, Ahmed AI, Shaman M, Lannaman K, Yoon BH, Hassan SS, Kim CJ, Korzeniewski SJ, Yeo L, Kim YM. 2015. Sterile and microbial-associated intra-amniotic inflammation in preterm prelabor rupture of membranes. The Journal of Maternal-Fetal and Neonatal Medicine 28:1394–1409. doi:10.3109/14767058.2014.958463

Wahid HH, Dorian CL, Chin PY, Hutchinson MR, Rice KC, Olson DM, Moldenhauer LM, Robertson SA. 2015. Toll-Like Receptor 4 Is an Essential Upstream Regulator of On-Time Parturition and Perinatal Viability in Mice. *Endocrinology* 156:3828–3841.